# Thermographic Image of the Hoof Print in Leisure and Cross-Country Warmblood Horses: A Pilot Study

**DOI:** 10.3390/vetsci10070470

**Published:** 2023-07-18

**Authors:** Cristian Zaha, Larisa Schuszler, Roxana Dascalu, Paula Nistor, Tiana Florea, Ciprian Rujescu, Bogdan Sicoe, Cornel Igna

**Affiliations:** 1Surgery Clinic, Faculty of Veterinary Medicine, University of Life Sciences “King Michael I”, 300645 Timisoara, Romania; larisaschuszler@usvt.ro (L.S.); roxanadascalu@usvt.ro (R.D.); nistorpaulanicoleta@yahoo.com (P.N.); 2Dermatology Department, Faculty of Veterinary Medicine, University of Life Sciences “King Mihai I”, 300645 Timisoara, Romania; tijana.florea@usvt.ro; 3Management and Rural Development Department, Faculty of Management and Rural Tourism, University of Life Sciences “King Michael I”, 300645 Timisoara, Romania; rujescu@usvt.ro; 4Diagnostic Imaging, Faculty of Veterinary Medicine, University of Life Sciences “King Michael I”, 300645 Timisoara, Romania; bogdan.sicoe@usvt.ro

**Keywords:** Warmblood horses, hoof print, thermography, reference temperature

## Abstract

**Simple Summary:**

The role of thermography as a detection method of the hoof print in non-lame Warmblood horses as well as its use in temperature determination in six areas from its surface was investigated in this pilot study. The study included sixty non-lame horses, and all four limbs of each horse were taken into consideration (*n* = 240). A comparison between the hoof print temperature values was performed between the horses used for leisure and those used for cross-country. The studied horses were selected based on the following criteria: no alterations in posture and no muscle group asymmetry during visual examination, no lateral or medial deviation of the carpus or hock, no reaction to the flexion tests, negative reactions to the hoof tester, no lameness during walking, trotting or lunging, no anti-inflammatory medication in the last three weeks and rectal temperature between 37 °C and 38 °C. The thermal patterns of the hoof print show no difference among the four limbs and the mean temperature of the selected areas presents no significant statistical differences. Also, there was no statistical differences between the mean temperature of the selected areas from the forelimbs and hindlimbs from the horses used for leisure and those used for cross-country. Thermography can detect the hoof print on a flat surface and the mean temperature for each studied area can be proposed as a reference temperature value. There were no differences in the mean hoofprint temperature between leisure and cross-country Warmblood Horses. Further investigations are required to clarify whether there are any differences in the thermal pattern of hoof prints from other breeds or from horses with musculoskeletal conditions.

**Abstract:**

Background: The field of veterinary medicine lacks information on equine thermal hoof printing, and few data on the same subject are available in dogs. In human medicine, thermography is used to detect heat emitted by the foot when it comes in contact with a flat surface to detect the abnormalities of the foot balance. The hypothesis states that the thermal pattern of the hoof print in Warmblood horses is detectable and it does not vary among the four limbs in leisure and cross-country Warmblood horses in terms of mean temperature of the hoof print surface. A pilot study was conducted to investigate the accuracy of thermography in temperature detection of the hoof print and to investigate the occurrence of possible differences in the mean value of six selected areas and whether there are any differences in the mean temperature of the hoof print between leisure and cross-country Warmblood horses. Methods: The study included sixty non-lame Warmblood horses with all limbs taken into consideration (*n* = 240). The selection criteria for the horses were: no alterations in posture and no muscle group asymmetry during visual examination, no lateral or medial deviation of the carpus or hock, no reaction to the flexion tests, negative reactions to the hoof tester, no lameness during walking, trotting or lunging, no anti-inflammatory medication in the last three weeks prior to examination and rectal temperature between 37 °C and 38 °C. The hoof print of each hoof was measured with the horse in the standing position, all four limbs on the ground, using a FLIR E50 thermal camera. Six areas of temperature from the hoof print were taken into consideration, and for each of them, the mean value was identified using FLIR Tools software for photo interpretation. The One-Way ANOVA test was used to test the differences between the mean temperatures obtained for each selected area from all limbs and to compare the hoof print temperature values between the leisure horses and cross-country horses. Data were statistically processed using SAS Studio. Results: Thermography can detect the temperature emitted by the hoof but the thermal patterns of the hoof print show no difference for all four studied limbs. No significant statistical differences were noticed between the mean temperatures identified for each studied area. Also, there were no statistical differences between the mean temperature of the selected areas from the forelimbs and hindlimbs from the horses used for leisure and those used for cross-country. Based on this aspect, the mean temperature of one selected area can be determined in any of the four limbs, without visible variations. Conclusions: Thermography can detect the hoof print on a flat surface and the mean temperature for each studied area can be proposed as a reference temperature value. There were no differences in the mean temperature of the hoofprint between leisure and cross-country Warmblood Horses. Further investigations are required to clarify whether there are any differences in the thermal pattern of hoof prints from other breeds or from horses with musculoskeletal conditions.

## 1. Introduction

The use of horses has changed over the last hundred years, moving from a central role in agriculture, industry and transportation to the new social assignment which is primarily represented by their use for sporting, leisure and recreational purposes [1]. Warmblood horses are used for sports such as dressage, show jumping and eventing all over the world. Differences among breeds and the effects on their learning ability prove that equine behavior is a heritable feature [2]. Cross-country jumping is a test of agility, endurance, and skill that implies following a predetermined course through fields and forests. Warmblood horses are an excellent choice for cross-country due to their large gallop strides, speed, and high endurance throughout long stretches of cantering or galloping [3]. A leisure horse, by definition, is used for activities that have no competitive elements, so these activities do not require the horse to exert intense effort in a short period of time. For both types of horses, the temperature of the hoof can be changed during exercises or in case of different pathologies that affect the blood supply [4,5].

Changes in temperature and blood flow can be visualized by thermal imaging. However, there is no information in veterinary medicine regarding the thermal hoof print in horses [5], but there is some information on the same subject in dogs [6,7]. Temperature differences of 0.5 °C were found in dogs between the paw print thermal image of the lame limb versus the non-lame limb, and these findings highlight the change in the thermal pattern of the paw print in the lame hindlimb compared to a non-lame hindlimb in both the lame and healthy groups [8]. The temperature differences between the paw prints aid the veterinarian in determining lameness, which can be translated into equine medical care. Due to a paucity of knowledge, we focused our pilot study on identifying the temperature of the hoof print in normal Warmblood horses.

Various methods of how to evaluate whether the horses are sound or show lameness are currently available; thus, the veterinarian can use clinical and lameness examination and can complete them with imaging scans (radiography, ultrasound, computed tomography—CT, magnetic nuclear resonance—MRI) [9,10,11]. All of these imaging evaluation methods require special and expensive equipment and physical or chemical manipulation of the horses [12,13]. Movement-based kinematic and kinetic gait tests are available to evaluate whether the horse is sound or suffers from lameness. Kinetic evaluation seeks to define and quantify the forces that produce a specific movement. Kinematics assess motion parameters both geographically (for example, height, displacement) and temporally (for example, duration, rate) [14,15]. There are drawbacks to adopting these approaches such as the limited range of allowed velocities which makes it difficult and time-consuming [14]. 

In contrast to radiography or computed tomography, the thermographic examination does not use any penetrating radiation and expensive devices. Also, the anatomical part that is examined does not receive X-rays, like in radiography; it is not placed in an electromagnetic field as in magnetic resonance imaging and does not use any radioactive substances, as in scintigraphy [12,16]. Thus, thermography scans can provide insight into inflammatory processes which affect blood flow to the feet [17]. It can show changes in temperature that may be associated with tendinitis, navicular syndrome, laminitis, sole abscesses and other hoof-related structural problems [18,19]. The thermographic scan provides an opportunity to discover subclinical inflammation in horse limbs 14 days before the clinical symptoms appear [20]. The advantages of using thermography are rapidity in the millisecond range and facilitating measurement of moving targets. Other advantages are provided by noncontact procedures, allowing measurements of hazardous or physically inaccessible objects, with no interference and no energy lost from the target [21,22,23]. There is no risk of contamination and no mechanical effect on the surface of the object [21,22,23].

Based on the information obtained from human medicine [24] and veterinary literature [5,6,7,8], the objectives of our study were as follows:to investigate the hoof print of non-lame horses;to calculate and propose a temperature reference value for six areas from the hoofprint surface;to check for any differences in temperature between the horses used for leisure and those trained for cross-country;to compare the results obtained in different locations.

The hypothesis was that the thermal pattern and the mean temperature obtained from the hoof print surface will not present differences among the four limbs in leisure and cross-country Warmblood horses.

## 2. Materials and Methods

The study was conducted in accordance with the guidelines of the Declaration of Helsinki, and approved by the Bioethics Commission of Banat University of Agricultural Sciences and Veterinary Medicine “King Michael I of Romania” from Timisoara with No. 51/07.06.2021.

### 2.1. Animal Selection

The study was performed in two private horse-riding centers (Location A and Location B) and in the Surgery Clinic of the Faculty of Veterinary Medicine from Timisoara (Location C) during the summer season of 2021. 

All horses were Warmblood horses: thirty-five of the horses were used for cross-country sports and fifty-two were used for leisure. The horses ranged in age from 4 to 16 years (mean 7.63 years and median 7.0 years), in weight from 510 to 615 kg (mean 545 kg) and height from 152 to 168 cm (mean 160 cm). 

There was no history of lameness throughout the last month prior to examination and no anti-inflammatory therapy within the last three weeks for all examined horses. The rectal temperature was between 37 and 38 °C for all horses and was measured at the beginning of the examination.

Before examination, the shoe was removed and the hooves were trimmed and balanced by a skilled qualified farrier with more than 15 years of experience of normal and therapeutic shoeing, and all the horses included in the study were trimmed by the same farrier. The hooves were trimmed according to barefoot trimming principles, which involved leveling the hoof to the live sole, lowering the heels and the frog, sole with bars remaining intact [25]. The normal angulation for the forelimbs was 50°, and for the hindlimbs it was 55°. The hoof was balanced in a medio-lateral (ML) shape and the coronet line was parallel with the ground surface and perpendicular to the line that bisects the limb axis when viewed from the front. All of the horses participated in the study with the consent of the owners. The horses brought into the clinic and those from the horse-riding centers were examined during routine check-ups. The thermographic examination was performed the following day after a routine check-up.

The inclusion criteria were as follows: no alterations in posture and no muscle group asymmetry at visual examination, no lateral or medial deviation of the carpus or hock, no reaction to the flexion tests, negative reactions to the hoof tester, and no lameness during walking, trotting and lunging. Measured rectal temperatures were between 37 °C and 38 °C. The exclusion criteria consisted in muscle group asymmetry, lameness during walking, trotting and lunging, positive reaction to the flexion tests, positive reaction to the hoof tester.

The horses were clinically examined following the agreement of two veterinarians with 11 and 15 years of experience in musculoskeletal injuries, with a lameness exam performed in accordance with the American Association of Equine Practitioners AAEP lameness system, with a 0 score for horses with no lameness under any circumstances and a 5 score for horses with non-weight-bearing lameness [26]. Medical history was obtained for each horse regarding the activity of the horse, most recent symptoms of lameness, previously received treatments or therapies. 

The horses were examined visually, during and through manipulative maneuvers and after the movement evaluation [9,10,11,27,28,29,30].

From the eighty-seven examined horses, sixty displayed soundness and twenty-seven presented different scores of lameness; Table 1 and Appendix A.

In those sixty horses selected for thermographic investigation, no vertical movements of the head or rotation of the pelvis were observed during lunging, and the horses moved symmetrically with equal movement of the head, trunk and limbs during walking, trotting or lunging on both the left and right sides; Appendix A.

### 2.2. Thermal Imaging and Data Recording

The intensity of infrared spectrum radiation in the hoof print (Figure 1) was evaluated after a one-hundred-second-long contact with a rubber floor surface to ensure a sufficient temperature transmission time from the hoof to floor surface and to not modify the weight bearing. The surface of the rubber was dry and without artifacts during examinations. Clayton and Nauwelaerts [31] investigated the pressure variables in non-lame horses during quiet standing on force plate for a period of sixty seconds. All four limbs were investigated separately, obtaining four hoof prints in the end. All the horses were subjected to the same procedure: the horses were maintained in the standing position, with all limbs in contact with the floor, shifting the position after one hundred seconds. The handler did not physically manipulate the horses during the recording period. The thermal scanning began with the left forelimb, followed by the right forelimb and continued with the left and right hindlimbs and was performed on the same day for each horse. The hoofprint was recorded using the Flir E50 camera in less than three seconds after the horse position was changed. 

Thermographic measurements were performed in the consultation room, at the Faculty of Veterinary Medicine and in the sheds of the riding horse centers.

All measurements for each studied horse were conducted observing the same environmental conditions for all examination spaces, i.e., dry, 12 mm-thick rubber floor at 18 °C, air temperature of 21 °C, humidity between 60% and 70%, and no air currents [32]. The floor of each examination room was flat and made of concrete over which a rubber material was applied. The measurements of the air temperature, humidity, and air currents were taken using the Testo 435 device. A period of 30 min was used for temperature acclimation of the horses in the examination room. The thermal images and measurements were recorded for each leg by the same non-blinded operator with nine years of thermography experience, at a distance of 50 cm at a 90° angle from the hoof print. Images were captured using a FLIR E50 (FLIR Systems Inc., Wilsonville, OR, USA) infrared camera set at 0.95 emissivity with a 240 × 180 resolution for each image. The temperature range was 20 to 650 °C, with a sensitivity of ≤0.05 °C. 

Six sections of the hoofprint were considered: El1—toe, El2 and El3—hoof wall, El4 and El5—sole, El6—frog apex, El7—frog, El8 and El9—heels. The mean temperatures for the toe (El1), frog apex (El6), and frog (El7) were taken into account, and for the remainder parts, the average between the mean areas was used, so for the hoof wall it was (El2 + El3/2), for the sole it was (El4 + El5/2) and for the heels it was (El8 + El9/2) (Figure 2). Each considered region was 5 × 3 pixels in size. We used the FLIR Tools software for picture interpretation to determine the mean temperatures of each area from the hoofprint, and the technique was carried out by the same operator. 

### 2.3. Statistical Analysis

For each temperature area from the thermal hoofprint of the hoof taken into study, the mean temperature was determined using the FLIR Tools software for photo interpretation.

The One-Way ANOVA test was used to test the differences between the mean temperature obtained for both the left and right hindlimbs, respectively, between the left and right hindlimbs, between left hindlimb and right hindlimb, right hindlimb and left hindlimb, as well as between all four limbs in all horses. Comparisons were made between the mean hoofprint temperatures of leisure and cross-country horses as well as between the locations where thermography was conducted. SAS Studio was used to carry out statistical analysis on the data, with statistical significance set at *p* > 0.05.

## 3. Results

### 3.1. Thermographic Scan of the Hoofprints

#### 3.1.1. Thermography of the Forelimbs

The hoof print was well defined on both the left and right limbs. There was an area of increased temperature in the region of the frog, the apex of the frog and the corresponding region of the hoof wall. The thermal shape of the frog was identifiable on the hoof print. The temperature of the sole’s thermal hoof print was consistent across its entire surface. The temperature in the heel area was less representative than in the other areas; Figure 3, Table 2.

#### 3.1.2. Thermography of the Hindlimbs

The hoof print was well defined on both hindlimbs. Areas of increased temperature were detected along the frog, frog apex and hoof wall. The temperature of the frog and its apex were identifiable, having the characteristic shape of the frog. The hoof wall temperature was higher than that of the other structures and could be identified on the edges of the hoof. The temperature of the sole hoof print was even across the entire surface. The temperature in the heel area was less detectable than in other areas; Figure 4, Table 3. 

The mean temperature of each area of the four limbs (*n* = 240) was calculated, and since there were no significant differences between the mean temperatures of the forelimbs and hindlimbs, we propose the mean temperature as a reference value for each studied area. For the toe area, the mean temperature obtained from the four limbs proposed as a reference value was 20.30 °C, with a 95% probability that this value lies in the range from 19.96 °C to 20.63 °C. For the sole area, the mean temperature obtained from the four limbs proposed as a reference value was 20.17 °C, with a 95% probability that this value lies in the range from 19.83 °C to 20.52 °C. For the frog area, the mean temperature obtained from the four limbs proposed as a reference value was 20.44 °C, with a 95% probability that this value lies in the range from 20.10 °C to 20.78 °C. For the frog apex area, the mean temperature obtained from the four limbs proposed as a reference value was 20.26 °C, with a 95% probability that this value lies in the range from 19.92 °C to 20.60 °C. For the hoof wall area, the mean temperature obtained from the four limbs proposed as a reference value was 20.12 °C, with a 95% probability that this value lies in the range from 19.77 °C to 20.46 °C. For the heel area, the mean temperature obtained from the four limbs proposed as a reference value was 20.01 °C, with a 95% probability that this value lies in the range from 19.66 °C to 20.35 °C.

### 3.2. Statistical Analysis—Group Comparisons

#### 3.2.1. Group Comparison Warmblood Horses

The One-Way ANOVA test revealed no statistically significant differences between the mean temperature measured for each studied location among the forelimbs and hindlimbs in all horses. The comparison includes the left and right forelimbs, as well as the left and right hindlimbs, the left forelimb and right hindlimb, and the right forelimb and left hindlimb, as well as the final comparison between all four limbs with *p* > 0.05 in all cases; Table 4. 

There were no statistically significant differences (*p* = 0.83) in mean toe temperature derived from the forelimbs and hindlimbs (*n* = 240). The boxplot in Figure 5 also depicts data series distributions for the four limbs, which are positioned in a similar relationship to one another.

In the case of mean sole temperature, there were no statistically significant differences (*p* = 0.89) between the means obtained (*n* = 240) from forelimbs and hindlimbs. The boxplot in Figure 6 also shows distributions of the data series for the four limbs positioned in a similar relationship to each other. 

In the case of mean frog temperature, there were no statistically significant differences (*p* = 0.73) between the means obtained (*n* = 240) from fore limbs and hindlimbs. The boxplot in Figure 7 also shows distributions of the data series for the four limbs positioned in a similar relationship to each other. 

In the case of the mean frog apex temperature, there were no statistically significant differences (*p* = 0.81) between the means obtained (*n* = 240) from forelimbs and hindlimbs. The boxplot in Figure 8 also shows distributions of the data series for the four limbs positioned in a similar relationship to each other. 

In the case of mean hoof wall temperature, there were no statistically significant differences (*p* = 0.81) between the means obtained (*n* = 240) from forelimbs and hindlimbs. The boxplot in Figure 9 also shows distributions of the data series for the four limbs positioned in a similar relationship to each other. 

In the case of mean heels temperature, there were no statistically significant differences (*p* = 0.86) between the means obtained (*n* = 240) from forelimbs and hindlimbs. The boxplot in Figure 10 also shows distributions of the data series for the four limbs positioned in a similar relationship to each other.

These results lead to the idea that the mean temperature of the toe, sole, frog, frog apex, hoof wall and heel areas can be determined in any of the four limbs, without certain limbs yielding different values in comparison to others.

#### 3.2.2. Group Comparisons 

Following the use of the One-Way ANOVA test, no statistically significant differences were found when comparing the mean temperatures obtained for each studied area among the forelimbs and hindlimbs between horses used for leisure and those used for cross-country and comparing the mean temperatures obtained for each studied area among the forelimbs and hindlimbs of horses between locations; Table 5, Appendix A. The comparison included all four limbs, with *p* > 0.05 in all situations.

## 4. Discussion

Thermography of the hoof print in horses in the standing position allowed us to check the temperature emitted by the hoof without lifting the limb. This thermal scanning of the hoof print may be an auxiliary method of hoof assessment performed in addition to the traditional orthopedic examination. Thermography results obtained from the forelimb and hindlimb revealed a similar thermal pattern in all limbs and no significant statistical differences between the mean of the areas taken into consideration. This confirmed the hypothesis regarding the mean temperature and thermal pattern of the hoof print stating that there are no differences among limbs. 

Thermography results obtained from the horses used for leisure and those used for cross-country showed no significant statistical differences between the mean temperature of the selected areas from the forelimbs and hind limbs and no changes in the thermal pattern between them in terms of temperature distribution on the surface of the hoofprint. The thermal pattern of the hoof print presents an area of increased temperature as a result of firm contact with the ground in the frog area, frog apex area, hoof wall, towards a uniform and reduced area of temperature in the sole and heels area where the contact with the ground is reduced. 

In our study, the result of the mean temperature of each selected area from the hoof print from both the left and right limb did not show significant differences based on the statistical test, so we proposed these temperatures as reference values for each area to additionally aid veterinarians when comparing with temperatures recorded in various pathologies. Similarly, results of tests performed on non-lame dogs revealed no differences between the left and right limbs compared with significant differences between left or right limbs of unilaterally lame dogs [7,8].

The room temperature during the investigation was 21 °C, a temperature that other authors recommend as having no influence on thermoregulation of the animals [32,33]. At low room temperatures, there is a decrease in blood flow in the distal parts, while in the case of room temperatures above the recommended range, vasodilation causes a generalized warming of the extremities, encouraging heat loss in the environment [33]. Xu et al. [29] reported that the thermal hoofprint offers more information when the environmental temperature is below 16 °C. In our study, the temperature of the rubber floor was 18 °C in order to facilitate the absorption of radiations emitted by hoofs with increased temperature. The mean temperature of the selected areas showed no differences between the hoofprint of horses which came from different investigation locations. The places where the thermographic investigation was carried out observed the same values of the environmental temperature as those of the temperature of the supporting surface. Also, no air currents were detected. All these preparations allowed the thermographic recording to benefit from identical conditions and no external influence so that the results would differ significantly between locations. 

An area of increased temperature was detected in the area of the frog, consequence of the distal arteries that supply the navicular area [6] where the blood supply can decrease as a result of thrombosis of the distal arterioles that supply the navicular bone, leading to pain and ischemic necrosis [34]. Along the hoof wall surface hoof print, the temperature distribution was different, showing areas of increased temperature probably due to the adaptation of blood vessels to the weight-bearing forces transferred to the microvasculature of the pododerma [35]. No areas of increased temperature were observed in the toe region considering the fact that in non-lame horses the support on the ground is achieved with the entire hoof and in palmar foot pain the support on the ground is sustained by the toe instead of the caudal part of the foot [33,36]. The temperature of the hoof can increase in case of hoof abscess when temperatures can reach values 6.17 °C higher compared to those of healthy hooves or in case of laminitis [17,37,38]. The temperature of the hoof can decrease in thrombosis, venous constriction of the laminar dermis, nerve damage and in case of therapeutic hypothermia [39,40,41].

In non-lame standing horses, the ground reaction forces can be measured using a force plate [42,43] were the forelimbs carry 58.6% of the body weight compared to 41.4% in the hindlimbs [44]; however, Back [15] found that sound Warmblood horses loaded their front limbs with 118% BW and their hindlimbs with 96% BW, with no significant differences between the forelimbs and hindlimbs. This fact indicates differences between forelimb and hindlimb forces during support, which, from the point of view of thermographic scanning, revealed no differences in temperature distribution or values. Further research is however required to correlate the pressure identified by the force plate with the thermal hoof print. A good support on the ground requires a balanced hoof that is symmetrical in size and shape and lands flat on the ground in order to optimize the efficiency and function of the foot [45,46]. In our study, the hooves were balanced and trimmed prior to thermal investigation and the thermal hoof print showed an equal distribution of the temperature along the hoof wall.

The study has several limitations, however; thus, we mention the moderate camera resolution of only 240 × 180 set at 0.95 emissivity for each image and the lack of ability to perform video sequences in the case of walking horses. A resolution for future studies implies using more performant cameras with resolutions higher than 640 × 480 pixels with the possibility of recording video sequences of the hoof print of the horses during walking. Another limitation was the failure to compare the results on different support surfaces and the fact that we did not use a force plate to assess the weight-bearing of each limb. Subsequent research will include correlation between the emitted force and the hoof print for each limb taken into study. The surface material consisted of rubber, but other materials (i.e., sponge, wood, sand) can be taken into account as well. For further studies, we recommend comparisons of hoof prints after hoof contact with several different support surfaces. The fact that we did not perform radiological assays nor did we evaluate the changes in serum biochemical parameters to establish the possibility of a metabolic disorder or chronic inflammatory disease as a cause of lameness is also a limitation. In further studies, we recommend blood tests to obtain the full blood count and biochemistry. Hoof conformation, namely thickness of the sole and frog in horses, represent another limitation. Thus, we recommend performing latero-medial radiographs in future studies to correctly measure the anatomical structures. The final limitation that we can take into consideration is represented by the contact time of the hoof with the floor surface, so future research must include a comparison between several periods of contact time of the hoof with the floor surface. Finally, comparisons among various contact periods between the hoof and the floor surface must be included in order to obtain a comprehensive view (i.e., 30 s, 60 s 150 s).

## 5. Conclusions

Thermography can detect the hoof print on a flat surface, and this study is the first investigation of the thermal hoof print in Warmblood horses. The mean temperature for each studied area of the hoof print did not vary between the forelimbs and hind limbs in leisure and cross-country Warmblood horses; thus, based on the results obtained in this pilot study, we propose the mean temperature of each selected area as a reference temperature value. Further investigations to clarify whether there are any differences in the thermal pattern of hoof prints from other breeds or from horses with musculoskeletal conditions are required.

## Figures and Tables

**Figure 1 vetsci-10-00470-f001:**
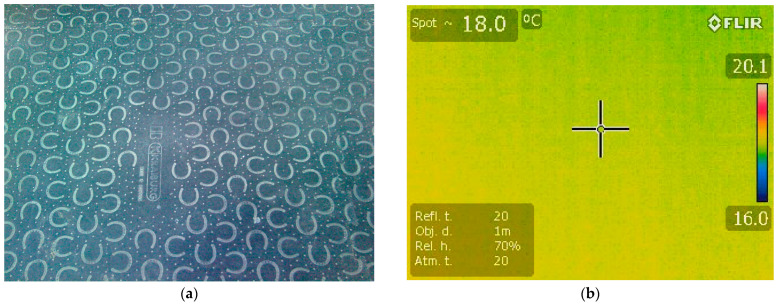
Photo of the floor surface: (**a**) normal image obtained with Flir E50; (**b**) thermographic image obtained with Flir E50; + (cross symbol) - spot of temperature.

**Figure 2 vetsci-10-00470-f002:**
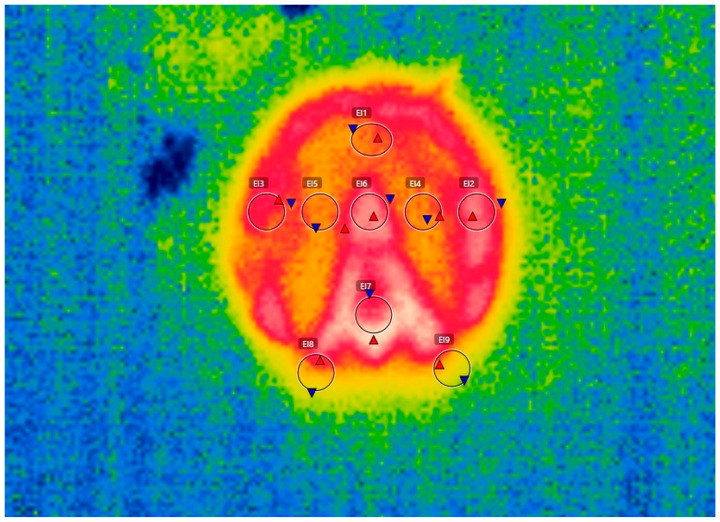
Hoofprint with marked areas: toe—El1, hoof wall—El2 and El3, sole—El4 and El5, frog apex—El6, frog—El7, heels—El8 and El9; minimum temperature recorded in the area—blue triangle; maximum temperature recorded in the area—red triangle.

**Figure 3 vetsci-10-00470-f003:**
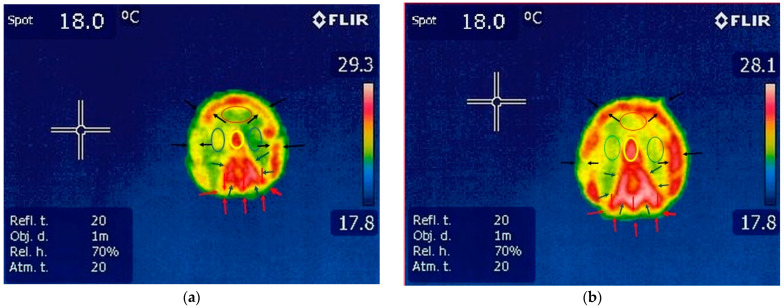
Thermography of the forelimbs: (**a**) left hoofprint: yellow circle—intense red color in frog apex corresponding with increased temperature; blue circles—sole areas with uniform color and temperature; red circle—uniform color for toe area with a decreased value of temperature; blue arrows—thermal aspect of the frog with red color which denotes increased temperature; red arrows—uniform color in heel area; black arrow—color variation in hoof wall areas with inequality in temperature; spot—temperature of the floor surface. (**b**) Right hoofprint: yellow circle—uniform red color corresponding with increased temperature in frog apex area; blue circles—uniform color for the sole area; red circle—uniform color in toe area; blue arrows—outline of the frog area with red color and increased value of temperature; black arrows—same color distribution among the hoof wall area; spot—temperature of the floor surface.

**Figure 4 vetsci-10-00470-f004:**
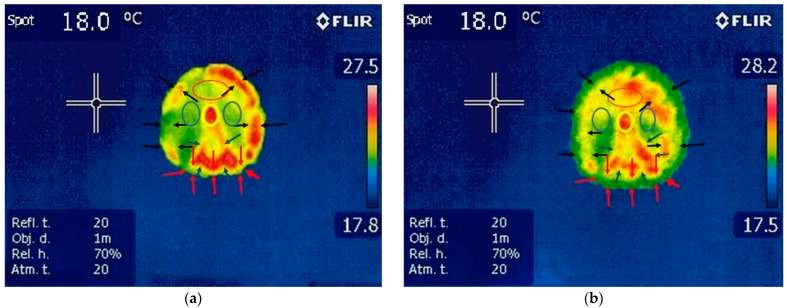
Thermography of the hindlimbs: (**a**) left hoofprint: yellow circle—uniform red color in frog apex corresponding with increased value of temperature; blue circles –uniform color and decreased value of temperature; red circle—uniform color for toe area with a decreased value of temperature; blue arrows—thermal aspect of the frog with red color which denotes increased temperature; red arrows—uniform color in heels area; black arrow—color variation in hoof wall areas with inequality in temperature; spot—temperature of the floor surface. (**b**) Right hooprint: yellow circle—uniform red color corresponding with increased temperature in frog apex area; blue circles—sole area with uniform thermal pattern; red circle—increased area of temperature in toe area with uniform color patter; blue arrows—outline of the frog area with red color and increased value of temperature; black arrows—same color distribution among the hoof wall area, red arrows—constant color distribution in heels area with decreased value of temperature; spot—temperature of the floor surface.

**Figure 5 vetsci-10-00470-f005:**
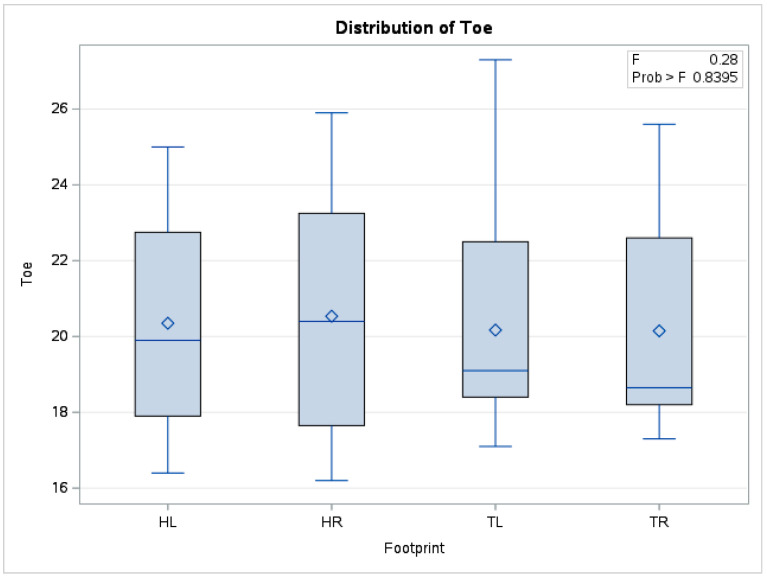
Comparative boxplot referring to results obtained from hoof print thermography for toe area; ◊—mean referee value; ―—median; HL—left hindlimb; HR—right hindlimb; TL—left forelimb; TR—right forelimb.

**Figure 6 vetsci-10-00470-f006:**
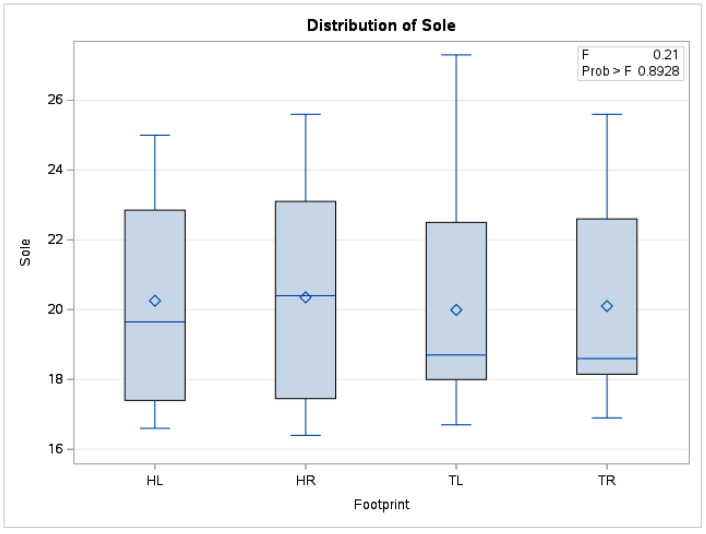
Comparative boxplot referring to results obtained from hoof print thermography for sole area; ◊—mean referee value; ―—median; HL—left hindlimb; HR—right hindlimb; TL—left forelimb; TR—right forelimb.

**Figure 7 vetsci-10-00470-f007:**
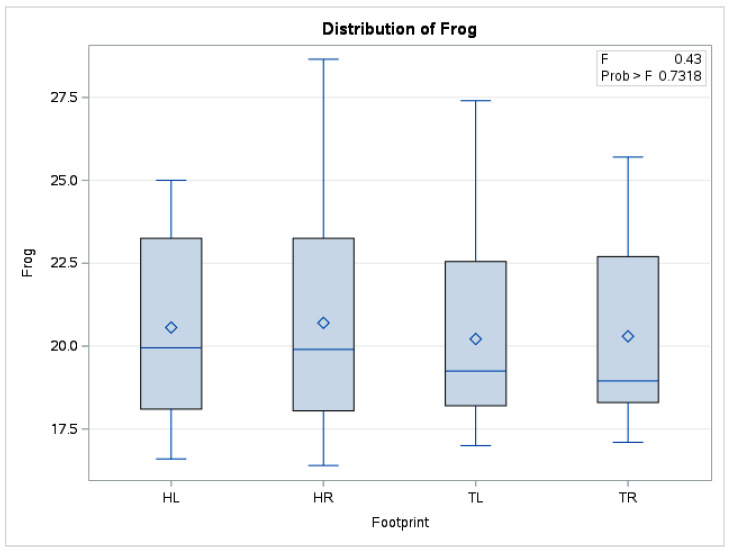
Comparative boxplot referring to results obtained from hoof print thermography for frog area; ◊—mean referee value; ―—median; HL—left hindlimb; HR—right hindlimb; TL—left forelimb; TR—right forelimb.

**Figure 8 vetsci-10-00470-f008:**
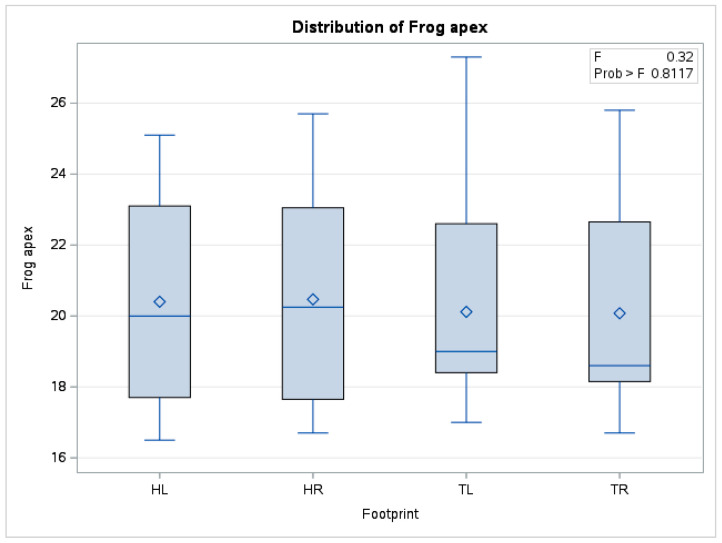
Comparative boxplot referring to results obtained from hoof print thermography for frog apex area; ◊—mean referee value; ―—median; HL—left hindlimb; HR—right hindlimb; TL—left forelimb; TR—right forelimb.

**Figure 9 vetsci-10-00470-f009:**
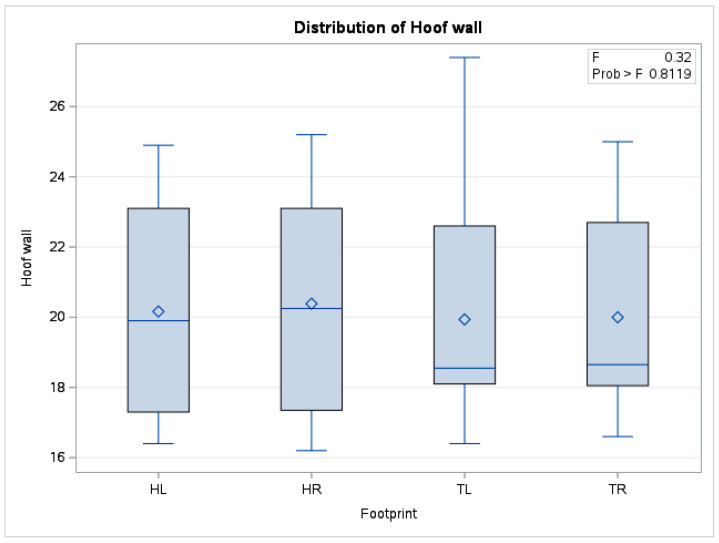
Comparative boxplot referring to results obtained from hoof print thermography for pentru hoof wall area; ◊—mean referee value; ―—median; HL—left hindlimb; HR—right hindlimb; TL—left forelimb; TR—right forelimb.

**Figure 10 vetsci-10-00470-f010:**
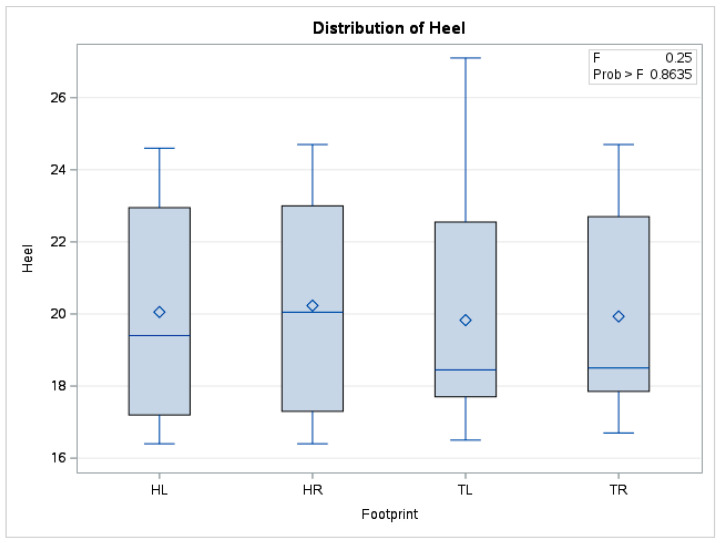
Comparative boxplot referring to results obtained from hoof print thermography for heels area; ◊—mean referee value; ―—median; HL—left hindlimb; HR—right hindlimb; TL—left forelimb; TR—right forelimb.

**Table 1 vetsci-10-00470-t001:** Results obtained after clinical and lameness examination.

Examination Method	Positive Response to Hoof Tester	Positive Reaction Joint Flexion	Vertical Movement of the Head during Walking	Rotation of the Pelvis	Gluteal Muscle and Sacral Tuberosities Asymmetries	Tendinitis of the Flexors	Alteration in Limb Positioning	Unique Condition	Multiple Condition
Number of affected horses	7	9	8	6	11	7	4	8	18

**Table 2 vetsci-10-00470-t002:** Descriptive statistics of the results obtained from the forelimbs hoof print thermography.

Position	Number	Variable	Mean	Std Dev	Minimum	Maximum	Median	Range	Lower 95% CL for Mean	Upper 95% CL for Mean
Forelimb (T)	120	Toe	20.44	2.72	16.20	25.90	20.25	9.70	19.94	20.93
		Sole	20.30	2.80	16.40	25.60	19.95	9.20	19.79	20.81
		Frog	20.63	2.79	16.40	28.65	19.90	12.25	20.12	21.13
		Frog apex	20.43	2.81	16.500	25.70	20.25	9.20	19.92	20.94
		Hoof wall	20.27	2.84	16.20	25.20	20.15	9.00	19.76	20.78
		Heel	20.14	2.79	16.40	24.70	19.65	8.30	19.63	20.64

**Table 3 vetsci-10-00470-t003:** Descriptive statistics of the results obtained from hindlimb hoof print thermography.

Position	Number	Variable	Mean	Std Dev	Minimum	Maximum	Median	Range	Lower 95% CL for Mean	Upper 95% CL for Mean
Hindlimb (H)	120	Toe	20.15	2.52	17.10	27.30	18.90	10.20	19.70	20.61
		Sole	20.04	2.62	16.70	27.30	18.60	10.60	19.57	20.52
		Frog	20.25	2.52	17.00	27.40	19.15	10.40	19.79	20.71
		Frog apex	20.09	2.58	16.70	27.30	18.70	10.60	19.63	20.56
		Hoof wall	19.96	2.63	16.40	27.40	18.60	11.00	19.49	20.44
		Heel	19.88	2.58	16.50	27.10	18.50	10.60	19.41	20.34

**Table 4 vetsci-10-00470-t004:** *p*-values—Group comparisons between left and right limbs, forelimbs and hindlimbs, and all limbs.

Group Comparison	Toe	Sole	Frog	Frog Apex	Hoff Wall	Heels
Left forelimb versus right forelimb	1	0.99	0.99	0.99	0.99	0.99
Left hindlimb versus right hindlimb	0.98	0.99	0.99	0.99	0.97	0.98
Left forelimb versus right hindlimb	0.87	0.88	0.75	0.89	0.80	0.84
Right forelimb versus left hindlimb	0.97	0.99	0.94	0.91	0.98	0.99
Forelimbs versus hindlimbs	0.83	0.89	0.73	0.81	0.81	0.86
All four limbs	0.60	0.70	1	0.81	0.61	0.58

**Table 5 vetsci-10-00470-t005:** *p*-values—group comparisons between leisure and cross-country hourses and between locations.

Comparison	Toe	Sole	Frog	Frog Apex	Hoof Wall	Heels
Leisure and cross-country horses	0.17	0.20	0.12	0.24	0.64	0.89
Location A, Location B and Location C	0.35	0.31	0.33	0.34	0.31	0.28

## Data Availability

All the data obtained for this pilot study are available in the Clinical Register of Large Animals of the Surgery Clinic.

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
