# Peer review of "Thermographic Image of the Hoof Print in Leisure and Cross-Country Warmblood Horses: A Pilot Study"

_vetsci, 2023, doi:10.3390/vetsci10070470_

Round 1
Reviewer 1 Report
Thank you for the manuscript. The study presented provides reference values for thermography of the foot in non-lame horses. Please see line by line revisions needed below:
Lay Summary and Abstract:
Minor grammatical errors present. Please revise
Introduction:
Line 59: Do not need to include statement about temperature being a measured quantity
Lines 64-67: Reword; very long sentence and difficult to follow
Line 71: Should this say “core body temperature”?
Line 77-78: Reword; sentence is confusing
Lines 80-85: Remove ultrasonography; this is also very non-invasive without any documented negative side effects
Line 87: “preventative”
Line 87-88: Is this proven? If not, reword such as “may have the ability”
Lines 93-96 : Expand on how thermography of human foot is useful. What were the results? Need an argument as to how this information justifies evaluation in the horse
Materials and Methods:
Put statistics section after all other described methods.
Line 125: AAEP grade 5/5 is non-weightbearing lameness
Lines 137-138: non-weightbearing
Were hooftesters, flexions, lameness performed before or after feet trimmed/shoes removed? For those evaluating horses, were grades assigned by consensus?
Lines 144-145: For hindlimb lameness, gluteal/pelvic excursion is usually assessed. Was this performed?
Statistics:
Was data normally distributed? If so, what test was used to evaluate?
Lines 168-176: How long did horses stand in room before thermography performed? Was there an acclimation period?
Lines 178-181: Please reword. Confusing.
Results:
Lines 191-192: Median listed only for ages
Line 194: Start sentence with “There was no history…”
Line 200: Say how many horses were excluded. Then list reasons for exclusion (lameness, hoof tester reactivity, positive flexion tests, etc)
Line 240: Include p-value
Line 256: “fore” limbs
Lines 260-301: Did you compare regions between left and right limbs? Different areas of the foot (sole and frog)? If so, this needs to be included (i.e. there was no significant difference between hoof wall area of the left and right pelvic limb (P-value =))
Discussion:
Please use either fore and hindlimb or thoracic and pelvic limb consistently throughout manuscript.
Lines 379-384: This is just repeating your materials and methods. Please remove or add in how this affected results, etc.
Lines 401-403: How do these facts relate to your results?
Please reorganize the discussion with the following recommendations:
1. Brief summary of findings and explain why your think this was found with evidence to support your thought process
2. How your results relate to findings in other species
3. How pathologies would change thermographic findings
4. Pros/cons of thermography versus other imaging modalities
5. Future studies
6. Limitations
Minor issues with grammar and language.
Author Response
Dear reviewer,
Thank you for your attention to giving us important information and new direction to improve our manuscript. We are grateful to your insightful comments.
Lay Summary and Abstract:
Minor grammatical errors present. Please revise
Response: we reviewed the abstract and corrected it.
Introduction:
Line 59: Do not need to include statement about temperature being a measured quantity
Response: We remove data about temperature and reorganize the introduction
Lines 64-67: Reword; very long sentence and difficult to follow
Response: We change in text from 118 to 122 line
Line 71: Should this say “core body temperature”?
Response: we remove this paragraph in the new introduction chapter
Line 77-78: Reword; sentence is confusing
Response: change in text at line 114 to 115.
Lines 80-85: Remove ultrasonography; this is also very non-invasive without any documented negative side effects
Response: remove in text and correct, radiography and computed tomography used penetrating radiation
Line 87: “preventative”
Response: remove the paragraph in the new introduction chapter
Line 87-88: Is this proven? If not, reword such as “may have the ability”
Response: remove the paragraph in the new introduction chapter
Lines 93-96 : Expand on how thermography of human foot is useful. What were the results? Need an argument as to how this information justifies evaluation in the horse
Response: add information in text at 147 to 150 lines.
Materials and Methods:
Put statistics section after all other described methods.
Response: change in text 2.3. Statistical analysis from 307 to 317.
Line 125: AAEP grade 5/5 is non-weightbearing lameness
Response: change in text at line 202.
Lines 137-138: non-weightbearing
Response: change in text at line 222.
Were hooftesters, flexions, lameness performed before or after feet trimmed/shoes removed? For those evaluating horses, were grades assigned by consensus?
Response: the examination was performed after shoe removal and hoof trimming at line 178. The lameness score were assigned by consensus after evaluation of two veterinarian, line 198.
Lines 144-145: For hindlimb lameness, gluteal/pelvic excursion is usually assessed. Was this performed?
Response: we didn’t performed gluteal/pelvic excursion, but we evaluate the gluteal muscle and sacral tuberosities asymmetries and manipulative joint tests (210 to 230 lines).
Statistics:
Was data normally distributed? If so, what test was used to evaluate?
Response: We did not perform the initial normality testing considering the fact that the total of the investigated samples constitutes a high volume sample. We thus choose for the use of parametric tests.
Lines 168-176: How long did horses stand in room before thermography performed? Was there an acclimation period?
Response: A period of 30 minutes was used for temperature acclimation of the horses in the examination room. Add in text at line 282-283
Lines 178-181: Please reword. Confusing.
Response: change at 295-303.
Results:
Lines 191-192: Median listed only for ages
Response: correct in text 170 -173
Line 194: Start sentence with “There was no history…”
Response: correct in text at line 174
Line 200: Say how many horses were excluded. Then list reasons for exclusion (lameness, hoof tester reactivity, positive flexion tests, etc)
Response: Twenty-seven horse were excluded, information at line 234-235. We add information in text (lines 234-244) and in supplementary tables Table S1 and Table S2.
Line 240: Include p-value
Response: a comparison to include p-value we add at the next subchapter. We present a qualitative assessment of the result and after the comparison.
Line 256: “fore” limbs
Response: at line 387 is “four limbs” (right and left forelimbs and right and left hindlimbs = total four limbs). Was made a comparison between areas taken into consideration of each limb.
Lines 260-301: Did you compare regions between left and right limbs? Different areas of the foot (sole and frog)? If so, this needs to be included (i.e. there was no significant difference between hoof wall area of the left and right pelvic limb (P-value =))
Response: We did comparison between left and right forelimb, between forelimbs and hindlimbs, and between all four limbs for each area took into consideration (i.e. comparison of mean temperature of the sole between each limbs) at line 456-525. Also comparison of the each area of temperature between each category of horses ( i.e. mean temperature of the sole from leisure horses from all limbs (n=144) with mean temperature of the sole from cross-country horses (n=96) at line 527-556.
Discussion:
Please use either fore and hindlimb or thoracic and pelvic limb consistently throughout manuscript.
Response: correct in the text used forelimbs and hindlimbs
Lines 379-384: This is just repeating your materials and methods. Please remove or add in how this affected results, etc.
Response: reorganize the discussion chapter. Remove these sentences
Lines 401-403: How do these facts relate to your results?
Response: at line 613-619. This fact indicates differences between thoracic and hind limbs forces during support, which from the point of view of thermographic scanning did not report differences in temperature distribution or values and requires future research to correlate the pressure identified by the force plate with the thermal hoof print
Please reorganize the discussion with the following recommendations:
- Brief summary of findings and explain why your think this was found with evidence to support your thought process
- How your results relate to findings in other species
- How pathologies would change thermographic findings
- Pros/cons of thermography versus other imaging modalities
- Future studies
- Limitations
Response: we manage the discussion section after your recommendation

Reviewer 2 Report
The work deals with a theme that is still current even if it is no longer prominent in clinical practice. Equinbe foot had not yet been addressed in such a specific way also because the technique had highlighted more than one diagnostic defect.
The work is a well-constructed pilot study with reliable data despite the use of older tools and software. The end result is that of having certain data on the region (horse's foot) but many doubts remain about the possible real use of the technique in clinical diagnostic practice in the face of the evolution of other much more performing techniques.
Author Response
Response:
Dear reviewer,
We thank you for your appreciations. In our pilot study we try to demonstrate the utility of thermography in scanning the hoof print in order to issue a referee temperature for each area taken into consideration in non-lame Warmblood horses. Also, our comparison check if there exist differences between hoof print temperature for Warmblood horses (n=240) and between horses used for leisure (n=144) and horses used for cross-country (n=96).
This thermal scanning of the hoof print may be an auxiliary method of hoof assessment performed in addition to the traditional orthopedic examination.

Reviewer 3 Report
While innovation associated with diagnostic tools utilized within the clinical setting can be instrumental for the equine veterinary industry, this manuscript falls short as it is repetitive throughout lacking the clarity, flow, organization, and objectiveness needed to be of value to the reader. It is unclear as to why the work is warranted, how the work was accomplished, and what impact it will have on the industry. The manuscript is disjointed with ideas not fully explored. A paragraph needs to be more than 1-2 sentences where a point is made and then it is fully explored and supported by valuable and relevant references. While it is a pilot study, and thus, the sample population is relatively small and a control is not present, the authors fall short in expanding on areas where there could be value in the findings, and this could be due to the structure and organization of the writing and/or the use of the data that was available.
To begin with, as a pilot study, the title needs to be focused on keeping to a more narrow lane in making claims about the findings. Of course, first, redirect the title from “footprint” to “hoofprint” as it is the hoof anatomical structures being examined as it relates to the thermal imaging. Furthermore, the sample population only covered Warmbloods, and thus, as a pilot study with a relatively small sample size, it is irresponsible to say that these findings were for all horses worldwide. Kinetic and kinematic data have reported differences between breeds so it is important to recognize these differences or to assume from previous published research that this might be an influencing factor until further work is done. Morphometric data along with force plates and accelerometer data have indicated the importance of results relevant to breed type. As a pilot study, not only is this acceptable, but also responsible to report as such. Similar research has also reported differences between performance type, and with half of the population divided into one type versus another performance type, it may not only be of value to look at potential differences between these performance types within this study, but to report responsibility within the title and throughout what performance types were utilized for this study. This leaves open possibilities for exploring potential variation within the equine population, and again, with only 60 horses utilized, that is a small sample population for making blanket statements for the entire horse population throughout the world. If the goal is to utilize this diagnostic tool for detecting subtle lamenesses, then, it is responsible to take a more detailed and focused approach.
The introduction is reflective of this issue concerning clarity for this manuscript. The first four paragraphs within the introduction are repetitive as this information later is repeated and these paragraphs lack direction and thorough development. Authors can start with the fifth paragraph for the introduction. Referring to line 96, expand on what was found for the dog information. Why was it so useful for clinicians and how would that be relevant to the equine clinician? Overall, the introduction needs to lay down a foundation for why the objectives and why the hypothesis. Specifically, why does this study focus on non-lame horses? Would a comparison between non-lame and a specific lameness be more applicable? Why are we concerned that there are no differences between the four limbs of the horse when it comes to thermal imaging? How is this information useful and why is it important? Although it may seem obvious to the authors, this needs to be clearly laid out to the reader with references that thoroughly support this justification and the directions of the study. While the authors focus their introduction more on why thermal imaging can be useful compared to other diagnostic tools, this study does not focus on comparing other tools to thermal imaging, but rather, comparisons made between the four limbs and the anatomical structures of the hoof when it comes to thermal imaging. This needs to be the focus of the introduction.
For the methods, while a statement is given at the end of the paper concerning protocol approval for the humane and ethical use of the animals for this study, this needs to be clearly given within the start of the methods. Consent is noted within both the methods and at the end, thus, the same needs to be done for this information. For the farrier trimming information, more exact and objective information including specific measurements needs to be included. The timing of the trimming as it relates to the timing of the thermal imaging for all of the horses utilized for this study needs to be made clear. Also, make clear as to whether the same farrier was used for all horses for this study. Furthermore, it is unusual that two locations were utilized for a pilot study, especially for a relatively small sample population. How many were taken from each site, and could only one site be utilized, since again this is a pilot study? If numbers are fairly split, variance analysis looking at impact on site needs to be explored to rule out this question concerning the possible influence of site. Finally, if both sites are left in, then, specific details concerning how the ground was similar enough so that it wouldn’t impact results must be given. For example, you have a shed versus a consultation room for thermal imaging between the two sites, and thus, under the rubber mat was it concrete and the same type of concrete for both sites? Even the preparation of concrete can influence the levelness of the ground surface utilized. How was it ensured both locations were equally level and was the same rubber mat used for all horses and for both sites? If used for all sites, how was the rubber mat cleaned and prepared for each thermal imaging session to ensure consistency throughout?
This study focused on sound horses, and thus, the sample population must be ensured that they are sound. Therefore, the soundness evaluations are not a part of the results. This is the inclusion/exclusion criteria before data collection. Move information accordingly. While not the results, it needs to be clearly understood and supported that the horses utilized were sound, especially when the goal is to study sound horses. While I agree with the authors that kinetic and kinematic data should have been utilized, particularly force plate measurements, radiographs would have given more insight on degenerative changes within the joints that could impact more subtle lamenesses. There is overwhelming data as to the limitation of the human eye, and thus, utilization of diagnostic tools that have been well studied in determining subtle lamenesses should have been utilized. In any case, the authors need to more thoroughly and objectively explain their assessments and what was observed and concluded from these assessments. A table showing the animals utilized after passing the exclusion criteria with specific details concerning age, height, weight, performance type, and assessment related results concerning health and soundness must be given. Also, justification as to why two veterinarians were utilized in the assessment needs to be given. Why not one or four? Were they both assessing all the horses, or did they divide up the animals assessed? Similar questions concerning justification for other aspects of the methods are present. In lines 172 and 184 it is noted that the same operator was used for these aspects of the methods of the study, but what qualifications do these individuals have to ensure what they did was carried out correctly? The success of thermal imaging similar to radiographs can be very dependent on the operators, and thus, this needs to be clear as to their level of expertise. In line 154, how was the length of 100 seconds determined to be acceptable? What previous research validates this length? Was the order of thermal imaging random for the four limbs? What was the order of thermal imaging and the time span that the horse may be standing for all of the imaging? Was it done all in one day?
As for the results, as mentioned previously, the information concerning the lameness evaluation is a part of the exclusion criteria and not a part of the results, and thus, should be moved accordingly. With this move, reduce language accordingly keeping more objective measures in place and utilize a table to indicate animals utilized for this study and their assessment results along with other details concerning the animals such as age, height, weight, performance type, etc. Be clear within the results, the final number of horses actually utilized for the study, not the ones excluded due to lameness issues. For the actual table given in the results concerning some of the thermal imaging data, report data dividing between left and right limbs, maybe move into two tables so that forelimbs given in one table and hind in the other. Give “n” specific to each limb, ie. right hind versus left hind. Include specific p-values for comparisons between the limbs, not just a blanket statement saying p > 0.05. Particularly in section 3.4 Group Comparisons, specific p values for all comparisons must be reported and this can be done within a table. Watch within the results to stay more objective in explanation of the data collected as lines 223-225 and 238-241 appears more subjective, qualitative assessment of results. This could be utilized within the discussion as results should focus more on the exact measurements taken. Analysis concerning variance between age, weight, height, performance type, and even location of data collection should be done and reported to rule out potential influence. If data was collected on different days throughout the summer or different times of the day, day and/or time may need to be a variable to consider just to rule out the influence of changing temperatures as the summer and/or day progresses. Lines 260-301 are quite redundant, and thus, don’t repeat what information is already available within a table. What information is given in these paragraphs easily can be reported within a table. Repeating every line as to the “N” is unnecessary and redundant, thus, adjust accordingly.
As for the discussion, similar to the introduction, this section is redundant and lacks clarity and organization. Lines 373-384 are redundant to previous sections and offer no additional information as to the value or understanding of the results reported. Lines 401-413 are choppy and lack development as to the points being made and how it relates to previous research. A paragraph longer than 1-2 sentences is necessary to truly develop an idea and support it through thorough investigation of previous research. In line 420, expand on why the authors recommend these “aims” and support accordingly why. Similarly, in lines 427-431, expand on why these are limitations and support accordingly and discuss how to best approach these limitations in future research. As a pilot study, limitations should be a large discussion point as to where to move from here and why despite the limits to this work this study is still relevant and useful. With better development, this will further support conclusions.
See comments and suggestions for further details concerning language.
Author Response
Dear Reviewer,
Thank you for your effort, attention, and new direction to improve our manuscript. We are grateful to your insightful comments.
Comments and Suggestions for Authors
While innovation associated with diagnostic tools utilized within the clinical setting can be instrumental for the equine veterinary industry, this manuscript falls short as it is repetitive throughout lacking the clarity, flow, organization, and objectiveness needed to be of value to the reader. It is unclear as to why the work is warranted, how the work was accomplished, and what impact it will have on the industry. The manuscript is disjointed with ideas not fully explored. A paragraph needs to be more than 1-2 sentences where a point is made and then it is fully explored and supported by valuable and relevant references. While it is a pilot study, and thus, the sample population is relatively small and a control is not present, the authors fall short in expanding on areas where there could be value in the findings, and this could be due to the structure and organization of the writing and/or the use of the data that was available.
Response: In our pilot study we try to demonstrate the utility of thermography in scanning the hoof print for Warmblood horse and we not used two categories of horse lame and non-lame that have to include a control group. Also, our comparison check if there exist differences between hoof print temperature for Warmblood horses (n=240) and a new direction between horses used for leisure (n=144) and horses used for cross-country (n=96). We hope that the new version of the manuscript it brings new data and is more concise.
To begin with, as a pilot study, the title needs to be focused on keeping to a more narrow lane in making claims about the findings. Of course, first, redirect the title from “footprint” to “hoofprint” as it is the hoof anatomical structures being examined as it relates to the thermal imaging. Furthermore, the sample population only covered Warmbloods, and thus, as a pilot study with a relatively small sample size, it is irresponsible to say that these findings were for all horses worldwide. Kinetic and kinematic data have reported differences between breeds so it is important to recognize these differences or to assume from previous published research that this might be an influencing factor until further work is done. Morphometric data along with force plates and accelerometer data have indicated the importance of results relevant to breed type. As a pilot study, not only is this acceptable, but also responsible to report as such. Similar research has also reported differences between performance type, and with half of the population divided into one type versus another performance type, it may not only be of value to look at potential differences between these performance types within this study, but to report responsibility within the title and throughout what performance types were utilized for this study. This leaves open possibilities for exploring potential variation within the equine population, and again, with only 60 horses utilized, that is a small sample population for making blanket statements for the entire horse population throughout the world. If the goal is to utilize this diagnostic tool for detecting subtle lamenesses, then, it is responsible to take a more detailed and focused approach.
Response: We agree with your observation and change the title to hoof print and the group of horse include the Warmblood ones and not the entire population as we notice in the previous title. A new comparison were made between the horses used for leisure (n=144) and those used for cross-country (n=96) for all mean temperature of the hoof print taken into consideration.
The introduction is reflective of this issue concerning clarity for this manuscript. The first four paragraphs within the introduction are repetitive as this information later is repeated and these paragraphs lack direction and thorough development. Authors can start with the fifth paragraph for the introduction. Referring to line 96, expand on what was found for the dog information. Why was it so useful for clinicians and how would that be relevant to the equine clinician? Overall, the introduction needs to lay down a foundation for why the objectives and why the hypothesis. Specifically, why does this study focus on non-lame horses? Would a comparison between non-lame and a specific lameness be more applicable? Why are we concerned that there are no differences between the four limbs of the horse when it comes to thermal imaging? How is this information useful and why is it important? Although it may seem obvious to the authors, this needs to be clearly laid out to the reader with references that thoroughly support this justification and the directions of the study. While the authors focus their introduction more on why thermal imaging can be useful compared to other diagnostic tools, this study does not focus on comparing other tools to thermal imaging, but rather, comparisons made between the four limbs and the anatomical structures of the hoof when it comes to thermal imaging. This needs to be the focus of the introduction.
Response: We change the introduction part according with your observation and start from the study that was performed in dogs regarding the paw print evaluation. The information obtained by other authors in dogs paw print investigation help the clinicians in identifying lameness, information that can be translated into horse medicine. In our pilot study, we focus to identify the temperature of the hoof print due to a lack of information for this field in horses, especially in Warmblood non-lame horses.
For the methods, while a statement is given at the end of the paper concerning protocol approval for the humane and ethical use of the animals for this study, this needs to be clearly given within the start of the methods. Consent is noted within both the methods and at the end, thus, the same needs to be done for this information. For the farrier trimming information, more exact and objective information including specific measurements needs to be included. The timing of the trimming as it relates to the timing of the thermal imaging for all of the horses utilized for this study needs to be made clear. Also, make clear as to whether the same farrier was used for all horses for this study. Furthermore, it is unusual that two locations were utilized for a pilot study, especially for a relatively small sample population. How many were taken from each site, and could only one site be utilized, since again this is a pilot study? If numbers are fairly split, variance analysis looking at impact on site needs to be explored to rule out this question concerning the possible influence of site. Finally, if both sites are left in, then, specific details concerning how the ground was similar enough so that it wouldn’t impact results must be given. For example, you have a shed versus a consultation room for thermal imaging between the two sites, and thus, under the rubber mat was it concrete and the same type of concrete for both sites? Even the preparation of concrete can influence the levelness of the ground surface utilized. How was it ensured both locations were equally level and was the same rubber mat used for all horses and for both sites? If used for all sites, how was the rubber mat cleaned and prepared for each thermal imaging session to ensure consistency throughout?
Response: At the beginning of the Material and Methods chapter we include the ethical statement approved by the Bioethics Commission of Banat University of Agricultural Sciences and Veterinary Medicine “King Michael I of Romania” from Timisoara with No.51/07.06.2021. The same farrier trimming all the horses and new data regarding the procedure was add in the text at line .. “The hooves were trimmed according to barefoot trimming principles, which involved leveling the hoof to the live sole, lowering the heels and the frog, sole with bars remaining intact [19]. The normal angulation for the forelimbs were 50° and for the hind limbs were 55°. The hoof was balanced in medio-lateral (ML) shape and the coronet line was parallel with the ground surface and perpendicular to the line that bisects the limb axis when viewed from the front.” We used different location because we did consultations in the clinic and external consultations on client place. The floor surface was the same for each place made from rubber and the constitution floor was made from concrete for each buildings. Also, we respect the environment and especially the temperature of the rubber floor during the examination. The surface of the rubber was dry and without artifacts. All the horses benefit from the same acclimation time for thermographic evaluation.
This study focused on sound horses, and thus, the sample population must be ensured that they are sound. Therefore, the soundness evaluations are not a part of the results. This is the inclusion/exclusion criteria before data collection. Move information accordingly. While not the results, it needs to be clearly understood and supported that the horses utilized were sound, especially when the goal is to study sound horses. While I agree with the authors that kinetic and kinematic data should have been utilized, particularly force plate measurements, radiographs would have given more insight on degenerative changes within the joints that could impact more subtle lamenesses. There is overwhelming data as to the limitation of the human eye, and thus, utilization of diagnostic tools that have been well studied in determining subtle lamenesses should have been utilized. In any case, the authors need to more thoroughly and objectively explain their assessments and what was observed and concluded from these assessments. A table showing the animals utilized after passing the exclusion criteria with specific details concerning age, height, weight, performance type, and assessment related results concerning health and soundness must be given. Also, justification as to why two veterinarians were utilized in the assessment needs to be given. Why not one or four? Were they both assessing all the horses, or did they divide up the animals assessed? Similar questions concerning justification for other aspects of the methods are present. In lines 172 and 184 it is noted that the same operator was used for these aspects of the methods of the study, but what qualifications do these individuals have to ensure what they did was carried out correctly? The success of thermal imaging similar to radiographs can be very dependent on the operators, and thus, this needs to be clear as to their level of expertise. In line 154, how was the length of 100 seconds determined to be acceptable? What previous research validates this length? Was the order of thermal imaging random for the four limbs? What was the order of thermal imaging and the time span that the horse may be standing for all of the imaging? Was it done all in one day?
Response: We move the information from the results section to Material and Methods in inclusion/exclusion section. We made table with the information of the horses include the age, weight, height, breed, lameness score. The horses were clinically examined after consensus of two veterinarians with 11 and 15 years of skills in musculoskeletal injuries with a lameness exam performed according to the American Association of Equine Practitioners AAEP lameness system. The examination was based by an traditional orthopedic examination include visual inspection and digital palpation of the anatomical structures. The lameness examination and the thermographic investigation was performed on two different days. The thermographic examination was performed by the same operator with an experience of nine years in thermographic examination at line 282. The time, 100 seconds in contact with the rubber surface we consider acceptable to ensure a sufficient temperature transmission time from the hoof to floor surface and to not modify the weight-bearing. At line 259, we didn’t found information regarding the specific time for hoof print thermography but Clayton and Nauwelaerts [28] investigate the pressure variables in non-lame horses during quiet standing on force plate for a period of sixty seconds. All the horses were subjected to the same procedure: the horses were maintained in standing position. The handler had no physical physical manipulation of the horse during the recording period. The thermal scanning began with the left forelimb, followed by the right forelimb and continued with the left and right hindlimbs and was performed on the same day for each horse.
As for the results, as mentioned previously, the information concerning the lameness evaluation is a part of the exclusion criteria and not a part of the results, and thus, should be moved accordingly. With this move, reduce language accordingly keeping more objective measures in place and utilize a table to indicate animals utilized for this study and their assessment results along with other details concerning the animals such as age, height, weight, performance type, etc. Be clear within the results, the final number of horses actually utilized for the study, not the ones excluded due to lameness issues. For the actual table given in the results concerning some of the thermal imaging data, report data dividing between left and right limbs, maybe move into two tables so that forelimbs given in one table and hind in the other. Give “n” specific to each limb, ie. right hind versus left hind. Include specific p-values for comparisons between the limbs, not just a blanket statement saying p > 0.05. Particularly in section 3.4 Group Comparisons, specific p values for all comparisons must be reported and this can be done within a table. Watch within the results to stay more objective in explanation of the data collected as lines 223-225 and 238-241 appears more subjective, qualitative assessment of results. This could be utilized within the discussion as results should focus more on the exact measurements taken. Analysis concerning variance between age, weight, height, performance type, and even location of data collection should be done and reported to rule out potential influence. If data was collected on different days throughout the summer or different times of the day, day and/or time may need to be a variable to consider just to rule out the influence of changing temperatures as the summer and/or day progresses. Lines 260-301 are quite redundant, and thus, don’t repeat what information is already available within a table. What information is given in these paragraphs easily can be reported within a table. Repeating every line as to the “N” is unnecessary and redundant, thus, adjust accordingly.
Response: The information regarding the horse lameness evaluation was moved in Material and Methods section. Two tables that includes age, weight, height, lameness and condition was composed of Table S1 and Table S2. Eighty-seven horses were evaluated and from these 27 were excluded after lameness evaluation and sixty were included in the study. At section Group comparison were compared data between left and right forelimbs, left and right hind limbs, all limbs of the Warmblood horses and data between leisure horse and those used for cross-country. Results of comparison with p –value was included into the text. At line 350-355 and 367-373 we add in text qualitative information. A table of p-value was included in text at line 469.
As for the discussion, similar to the introduction, this section is redundant and lacks clarity and organization. Lines 373-384 are redundant to previous sections and offer no additional information as to the value or understanding of the results reported. Lines 401-413 are choppy and lack development as to the points being made and how it relates to previous research. A paragraph longer than 1-2 sentences is necessary to truly develop an idea and support it through thorough investigation of previous research. In line 420, expand on why the authors recommend these “aims” and support accordingly why. Similarly, in lines 427-431, expand on why these are limitations and support accordingly and discuss how to best approach these limitations in future research. As a pilot study, limitations should be a large discussion point as to where to move from here and why despite the limits to this work this study is still relevant and useful. With better development, this will further support conclusions.
Response: The Discussion chapter was modified and brings new information to understand the obtained results. We delete the information from 373-384. We complete the further research and direction of this pilot study starting from line 642-664. The limitation of the study we detailed them from 665 to 689 lines.

Reviewer 4 Report
An interesting article, presenting the method of thermographic analysis of the body surface for the diagnosis of disease changes, already known in veterinary medicine.
The authors decided to develop reference ranges of hoof temperature in healthy horses.
The question is - why?
Will this be an auxiliary method of hoof assessment performed in addition to the traditional orthopedic examination?
Will this be an amateur hoof evaluation method by the owner?
Will this be a mandatory procedure for sport horses?
Will it apply to draft horses?
The "Animal information" section regarding the description of animals should in my opinion be in the "Material and Methods" chapter and not in "Results"
What about ultrasound examination of tendons and tendon sheaths - was it or not?
I am concerned that external examination cannot rule out chronic inflammatory changes that may affect blood circulation in the horse's distal toe.
What about the study of basic parameters of whole blood and biochemical parameters of blood serum? They are necessary to recognize that the animal is healthy.
Author Response
Dear reviewer,
We thank you for your attention to read and give us information and new direction to improve our manuscript.
Comments and Suggestions for Authors
An interesting article, presenting the method of thermographic analysis of the body surface for the diagnosis of disease changes, already known in veterinary medicine.
The authors decided to develop reference ranges of hoof temperature in healthy horses.
The question is - why?
Will this be an auxiliary method of hoof assessment performed in addition to the traditional orthopedic examination?
Will this be an amateur hoof evaluation method by the owner?
Will this be a mandatory procedure for sport horses?
Will it apply to draft horses?
Response: In our pilot study, we wanted to point out the usage of thermography to detect the hoo print in non-lame Warmblood horses. We demonstrate that there are no significant differences between the mean temperature of each area took into consideration from hoof print of each limb (n=240) or between the mean temperature of each hoof print from the leisure horses comparative with the cross-country horses. Based on this aspect, the mean temperature of one selected area can be determined in any of the four limbs, without visible variations and didn’t require lifting the limb to detect the temperature. This pilot study can represent the starting point for comparing the thermal hoof print from horses with various pathologies and can be an auxiliary method of hoof assessment in addition to the traditional orthopedic examination.
The "Animal information" section regarding the description of animals should in my opinion be in the "Material and Methods" chapter and not in "Results"
Response: we change in text and move the Animal information from “Results” to “Material and Methods”
What about ultrasound examination of tendons and tendon sheaths - was it or not?
Response: there was performed visual inspection and digital palpation of the tendon and tendon sheaths.
I am concerned that external examination cannot rule out chronic inflammatory changes that may affect blood circulation in the horse's distal toe.
What about the study of basic parameters of whole blood and biochemical parameters of blood serum? They are necessary to recognize that the animal is healthy.
Response: We based on the orthopedic examination and we didn’t perform the biochemical parameters. We think the positive reaction to the test give us information to distinguish the non-lame versus lame horses.

Round 2
Reviewer 1 Report
Manuscript is much improved compared to original submission. Detailed revision recommendations are below:
Line 38: "The hypothesis was that thermal pattern..."
Line 52: "compare" instead of "comparison"
Line 74-75: "temperature of normal Warmblood horses"
Line 87: "In one study, thermographic scan..."
Line 107: "patients"
Line 109: Reword. Perhaps something like: "Thermographic evaluation of hoof print may therefore be useful in the horse to evaluate different musculoskeletal diseases or imbalance of the hoof"
Line 118-119: Add in hypothesis of leisure vs cross-country horses
Statistical Analysis: Did you test the data for normality before using one-way ANOVA?
Line 402: Please use either fore or thoracic and hind or pelvic terms consistently throughout manuscript
Line 439: "where"
Line 462: Do you have evidence that thermography can be used preventatively?
Line 464-465: Lack of making a specific diagnosis is a disadvantage. Please state clearly.
Line 469: What about differences for other breeds? Please clarify that these references can only be applied to Warmbloods throughout the manuscript
English is good with only minor changes required.
Author Response
Resposes to reviewer 1
We bring thanks to the reviewer for his time to reviewing the manuscript and for his observation regarding our work. We answer to each question and update the change in text using Track Change.
Manuscript is much improved compared to original submission. Detailed revision recommendations are below:
Line 38: "The hypothesis was that thermal pattern..."
Response: updated in text at line 41
Line 52: "compare" instead of "comparison"
Response: we have fixed the error at line 62.
Line 74-75: "temperature of normal Warmblood horses"
Response: we update in text at line 103.
Line 87: "In one study, thermographic scan..."
Response: we correct in text at line 134.
Line 107: "patients"
Response: we have fixed the error at line 184.
Line 109: Reword. Perhaps something like: "Thermographic evaluation of hoof print may therefore be useful in the horse to evaluate different musculoskeletal diseases or imbalance of the hoof"
Response: we agree and have updated at line 186.
Line 118-119: Add in hypothesis of leisure vs cross-country horses.
Response: add in text “…..will not present differences in all four limbs for leisure and cross-country Warmblood horses.”
Response: the observation is correct. We have changed in text at line 198.
Statistical Analysis: Did you test the data for normality before using one-way ANOVA?
Response: Upon your suggestion, this time we included the indicators of a normality test, namely the Shapiro-Wilk test. They describe the movement of the obtained series compared to the normal distribution. However, a data set of this type could be part of a normally distributed population. Extreme temperature values should be rarer frequency compared to the average temperatures. We cannot explain the displacement of the obtained series compared to the normality curve. Considering these aspects and the volume of analyzed data, in the end we chose to use parametric tests.
Line 402: Please use either fore or thoracic and hind or pelvic terms consistently throughout manuscript
Response: we correct in text in forelimb and hindlimb.
Line 439: "where"
Response: we correct in text
Line 462: Do you have evidence that thermography can be used preventatively?
Response: Yes, Soroko and Howell [1] exclaim in conclusions: “Thermography can play an important complementary role in the early detection and treatment of pathology and help to prevent pathologic conditions and the financial losses associated with delayed diagnosis “. In the same article by Soroko and Howell [1]: “Thermography can protect the horse by detecting signs of inflammation in the distal parts of the limbs before any clinical evidence such as lameness is present. This allows training programs to be adapted accordingly to aid injury prevention”.
Line 464-465: Lack of making a specific diagnosis is a disadvantage. Please state clearly.
Response: change in text at line 690.
Line 469: What about differences for other breeds? Please clarify that these references can only be applied to Warmbloods throughout the manuscript
Response: The observations were made only for Warmblood horses, we updated in the manuscript.

Reviewer 3 Report
Authors should be commended on the extent of the revisions made within a short amount of time and the additional supplementary material provided to support research methods. However, in the process of updating the manuscript, it is obvious that the revisions were rushed as the disjointed nature of the writing and the lack of flow and organization have become worst from that of the original submission. There are multiple grammatical errors along with poor sentence and paragraph structuring. Transitioning between paragraphs are poor within the introduction and discussion making it hard to follow at times. Finally, there are some serious issues with the added tables including the supplementary materials on formatting with information missing concerning what is being reported.
The sample size is still relatively low to represent all Warmblood horses. Was any power analysis done to determine proper sample size? Warmblood may need to be narrowed down to the specific type evaluated within this study. Since comparisons were made between performance type, this too needs to be indicated within the title and further explained within the introduction. Authors can, however, since there were no differences in what was reported within the results concerning performance type, just indicate this information in the methods as to no differences despite different type of performance types, and thus, data was reported as one group. Then, just include that information concerning these comparisons within a table to show no differences. If, however, the authors want to address the differences between performance types as this could be of interest to readers, then, within the results in lines 370-397 move that information to a table. In addition, further exploration of this discussion point concerning performance types within the introduction, discussion, and conclusions should be done. This comparison should also be reflected in the title.
More objective diagnostic testing should have been done on these animals prior to testing beyond visual observations to ensure subtle unsoundness issues were not present. Unfortunately, this is not something that can be rectified through revisions without redoing the study. Authors are commended, however, on their detailed description of the clinical examination, but that is unnecessary as it follows typical lameness examination procedures. A simple reference to a publication that reflects these procedures can be done instead to reduce this section within the methods. In fact, authors can reduce this section within the methods to just lines 156-162, and then, include a table with the results of the examination.
Along with a lack of a control for the study, another area that cannot be corrected after the fact is the use of two different sites, however, the authors do a good job in expanding on the conditions criteria, although statistical comparisons of the sites to ensure there were not variations between subjects that could be attributed to sites should be done. Data can be included within the supplementary material with authors noting no differences within this section of the methods. Although if differences are found, this could be addressed by utilizing one location for the study and removing the other site's data. This will, however, reduce sample size, which is already a problem, but will help to minimize potential variables influencing results. Additional revisions include lines 287-308, as this is redundant from what is given within the table, and thus, can be reduced to avoid redundancy.
As for the discussion, with suggested changes given above and the issues currently present as to flow and organization, this section needs to be more thoroughly revised. As mentioned within the previous reviewer comments, the short 2-3 sentence paragraphs do not allow for a thought to be thoroughly addressed and explored. While limitations were added to the discussion, these were not thoroughly explored either concerning ways they were addressed within this study and how in the future they could be further addressed. Finally, the conclusions can be more streamlined focusing on the main points and application to the veterinary industry.
As for a concluding note to authors concerning this manuscript, despite the issues concerning the revisions needed and the current limitations, it appears to be a good amount of data available and additional comparisons that are possible, thus, authors might consider forming two manuscripts from the current one. This would, of course, require more statistical analysis concerning the limitation related to sample size, but there is potential. Even if power analysis is poor, this information could be presented as a pilot study. In the end, the disjointed nature and lack of organization within the current manuscript could be more due to the amount of data available in which this could be addressed simply by dividing within multiple manuscripts or addressing some of the concerns as given above. Either direction will take time to address properly to give true credit to the work that has already been done.
See comments above. Authors need outside assistance on writing style that can meet publication standards. Data holds promise, but is poorly presented due to poor techniques in writing style.
Author Response
Response to Reviewer 3:
We want to thank the reviewer for his time spent carefully reviewing the manuscript and in their opinions regarding the material.
Authors should be commended on the extent of the revisions made within a short amount of time and the additional supplementary material provided to support research methods. However, in the process of updating the manuscript, it is obvious that the revisions were rushed as the disjointed nature of the writing and the lack of flow and organization have become worst from that of the original submission. There are multiple grammatical errors along with poor sentence and paragraph structuring. Transitioning between paragraphs are poor within the introduction and discussion making it hard to follow at times. Finally, there are some serious issues with the added tables including the supplementary materials on formatting with information missing concerning what is being reported.
Response: We bring changes to the manuscript regarding the fluent description of the manuscript. We updated the paragraph in the introduction and discussion. New information was added in the supplementary material.
The sample size is still relatively low to represent all Warmblood horses. Was any power analysis done to determine proper sample size? Warmblood may need to be narrowed down to the specific type evaluated within this study. Since comparisons were made between performance type, this too needs to be indicated within the title and further explained within the introduction. Authors can, however, since there were no differences in what was reported within the results concerning performance type, just indicate this information in the methods as to no differences despite different type of performance types, and thus, data was reported as one group. Then, just include that information concerning these comparisons within a table to show no differences. If, however, the authors want to address the differences between performance types as this could be of interest to readers, then, within the results in lines 370-397 move that information to a table. In addition, further exploration of this discussion point concerning performance types within the introduction, discussion, and conclusions should be done. This comparison should also be reflected in the title.
Response: We understand that the confidence level would rise significantly if the number of tested animals was also higher, but, in agreement with our testing capacity, this operation would require an extended period of time, which is one of the reasons why we chose to do a pilot study. We agree and update the title including the comparison between the leisure and cross-country Warmblood horses. We include information regarding the result of the comparison between leisure and cross-country Warmblood horses into Table 6 (line 549) and supplementary material for each area of the hoofprint regarding the comparison in the Figure S1- Figure S6. Information regarding the data for performance type were add in introduction, discussion and conclusions.
More objective diagnostic testing should have been done on these animals prior to testing beyond visual observations to ensure subtle unsoundness issues were not present. Unfortunately, this is not something that can be rectified through revisions without redoing the study. Authors are commended, however, on their detailed description of the clinical examination, but that is unnecessary as it follows typical lameness examination procedures. A simple reference to a publication that reflects these procedures can be done instead to reduce this section within the methods. In fact, authors can reduce this section within the methods to just lines 156-162, and then, include a table with the results of the examination.
Response: We are agree with the comment regarding the diagnosis methods and believed that the clinical, manipulative and movement examination give important information for diagnosis of soundness or lameness. We reduce the Materials and Methods section and include a Table 1 (line 247) with the result of the examination.
Along with a lack of a control for the study, another area that cannot be corrected after the fact is the use of two different sites, however, the authors do a good job in expanding on the conditions criteria, although statistical comparisons of the sites to ensure there were not variations between subjects that could be attributed to sites should be done. Data can be included within the supplementary material with authors noting no differences within this section of the methods. Although if differences are found, this could be addressed by utilizing one location for the study and removing the other site's data. This will, however, reduce sample size, which is already a problem, but will help to minimize potential variables influencing results. Additional revisions include lines 287-308, as this is redundant from what is given within the table, and thus, can be reduced to avoid redundancy.
Response: We reorganize the comparison area and add a new subcapitol 3.2.3. Group comparison between location (line 551-556) and the obtained result showed no differences between the location. The results of p-value for each comparisons are in Table 7 and Supplementary data in Figure S7-S12.
As for the discussion, with suggested changes given above and the issues currently present as to flow and organization, this section needs to be more thoroughly revised. As mentioned within the previous reviewer comments, the short 2-3 sentence paragraphs do not allow for a thought to be thoroughly addressed and explored. While limitations were added to the discussion, these were not thoroughly explored either concerning ways they were addressed within this study and how in the future they could be further addressed. Finally, the conclusions can be more streamlined focusing on the main points and application to the veterinary industry.
Response: We reorganize the discussion section and add information for limitation and further investigations. The conclusion section was updated as well.
As for a concluding note to authors concerning this manuscript, despite the issues concerning the revisions needed and the current limitations, it appears to be a good amount of data available and additional comparisons that are possible, thus, authors might consider forming two manuscripts from the current one. This would, of course, require more statistical analysis concerning the limitation related to sample size, but there is potential. Even if power analysis is poor, this information could be presented as a pilot study. In the end, the disjointed nature and lack of organization within the current manuscript could be more due to the amount of data available in which this could be addressed simply by dividing within multiple manuscripts or addressing some of the concerns as given above. Either direction will take time to address properly to give true credit to the work that has already been done.
Response: We are grateful for the appreciation brought to our work and we suppose after your insightful suggestion regarding the comparison between locations the result of the statistical analysis was positive with no differences. Our work presents a pilot study for the hoof print in horses and certainly requires a further comparison between non-lame and lame horses and between different breeds of horses.

Round 3
Reviewer 3 Report
Authors are commended for their continued work and revisions associated with their manuscript. Their dedication to improving their manuscript shows their belief in the value of the science being presented within this manuscript. While there are some serious flaws in the research methods utilized for this study, the authors recognize the limitations and present accordingly as a pilot study. Hopefully, to provide merit to the work that is done within this pilot study, they will take notice of these limitations within future projects.
To present this work within a scientific journal, there are, however, a few areas that continue to need improvement, which will allow for flow and clarity of thought. In the introduction, in line 89, change "doctor" to "veterinarian". Prior to the sentence in line 83, which is also the start of a new paragraph, add the following: "Changes in temperature and blood flow can be visualized by thermal imaging. However, .....", then, continue on with your sentence in line 83 as this will assist in transitioning from the previous paragraph. Change verb tense in line 93 so that it says "the veterinarian can use clinical and lameness examination and can complete them with imaging scans....", then, continue on with that sentence. Remove lines 103-104 as more useful in the discussion section. Combine paragraph ending on line 110 with the paragraph starting on line 111. Use present tense verb in line 114 so that it says "provides" instead of "provided". Take out lines 121-126 as this information questions why you didn't do a complementary approach with radiographs and ultrasonography for your study. Take out lines 126-132 as this information is not needed. Make sure after all the revisions to check references to update for any references that aren't cited due to removal of information.
Within the methods, move tables and figures right after text that introduces them. For example, move Table 2 just after Figure 3. Move Table 3 just after Figure 4. Remove Table 4 as information is repeated in the text. Combine into one paragraph lines 292-313. Combine sections 3.2.2. and 3.2.3. and make into one section and label as "Group Comparisons ". Then, combine Tables 6 and 7 into one table and make sure to change the commas to periods within the p-values reported. Remember since removing Table 4 and combining 6 and 7 to renumber tables accordingly including within the text.
As for the discussion, combine paragraphs that are located within lines 401-408 as similar in nature. Same for combining paragraphs within lines 424-438 along with combining paragraphs within lines 439-449. This will improve the choppiness within this section. Remove lines 450-458 as this is redundant from the introduction. Remove lines 459-482 as the results along with previous and following discussion points will suggest areas for future studies, and thus, just is redundant information. Combine paragraphs within lines 483-505. As for the conclusions section, similarly, combine lines 506-515 into one paragraph.
See previous comments.
Author Response
Authors are commended for their continued work and revisions associated with their manuscript. Their dedication to improving their manuscript shows their belief in the value of the science being presented within this manuscript. While there are some serious flaws in the research methods utilized for this study, the authors recognize the limitations and present accordingly as a pilot study. Hopefully, to provide merit to the work that is done within this pilot study, they will take notice of these limitations within future projects.
To present this work within a scientific journal, there are, however, a few areas that continue to need improvement, which will allow for flow and clarity of thought. In the introduction, in line 89, change "doctor" to "veterinarian". Prior to the sentence in line 83, which is also the start of a new paragraph, add the following: "Changes in temperature and blood flow can be visualized by thermal imaging. However, .....", then, continue on with your sentence in line 83 as this will assist in transitioning from the previous paragraph. Change verb tense in line 93 so that it says "the veterinarian can use clinical and lameness examination and can complete them with imaging scans....", then, continue on with that sentence. Remove lines 103-104 as more useful in the discussion section. Combine paragraph ending on line 110 with the paragraph starting on line 111. Use present tense verb in line 114 so that it says "provides" instead of "provided". Take out lines 121-126 as this information questions why you didn't do a complementary approach with radiographs and ultrasonography for your study. Take out lines 126-132 as this information is not needed. Make sure after all the revisions to check references to update for any references that aren't cited due to removal of information.
Within the methods, move tables and figures right after text that introduces them. For example, move Table 2 just after Figure 3. Move Table 3 just after Figure 4. Remove Table 4 as information is repeated in the text. Combine into one paragraph lines 292-313. Combine sections 3.2.2. and 3.2.3. and make into one section and label as "Group Comparisons ". Then, combine Tables 6 and 7 into one table and make sure to change the commas to periods within the p-values reported. Remember since removing Table 4 and combining 6 and 7 to renumber tables accordingly including within the text.
As for the discussion, combine paragraphs that are located within lines 401-408 as similar in nature. Same for combining paragraphs within lines 424-438 along with combining paragraphs within lines 439-449. This will improve the choppiness within this section. Remove lines 450-458 as this is redundant from the introduction. Remove lines 459-482 as the results along with previous and following discussion points will suggest areas for future studies, and thus, just is redundant information. Combine paragraphs within lines 483-505. As for the conclusions section, similarly, combine lines 506-515 into one paragraph.
Response to Reviewer 3:
We bring thanks to the reviewer for his time to assess the manuscript and for his observation regarding our work. Our pilot study represents a start for hoofprint thermography in Warmblood horses used for leisure and cross-country and will require further investigation of the horse hoofprint in which it will be followed differences between non-lame and lame horses, and definitely we will work on the limitations and improve the methods section.
Responses according to the observations:
Introduction section:
· Line 89: We’ve corrected in text with veterinarian, at line 90
· Line 83: We’ve made the change at line 83. “Changes in temperature and blood flow can be visualized by thermal imaging. However, there is no…”
· Line 93: We’ve corrected in text the verb tense at line 94: “…the veterinarian can use clinical and lameness examination and can complete them with imaging scans…”
· Line 103-104: We change the information and insert it in Discussion section at line 416-417.
· Line 110-110: We’ve correct and combine the line from 110 and 111 at line 107-110
· Line 114: We’ve corrected in text with “provides” at line 112
Results section:
We change in text and add the Table 2 after Figure 3 (Line 259) and Table 3 after Figure 4 (Line 280) and remove the Table 4. We combine in one paragraph the lines 292-313 with lines 282-297. We combine the section 3.2.2. and 3.2.3 in one section (3.2.2. Group comparisons – Line 352) and we combine also the Tables with p-value in one as Table 5 (Line 360). After the changes, we took care to respect their numbering.
Discussion section:
· Line 401-408: we change in text at lines 378-383
· Line 424-438: We’ve made the change in text at lines 400-412
· Line 439-449: We change in text at lines 414-425
· Line 450-458: We remove the information
· Line 459-482: We remove the information
· Line 483-505: We’ve made the change at lines 427-449
Conlusions:
Line 506-515: we change in text and combine into one paragraph on lines 451-458.
